# Synthesis and Initial In Vivo Evaluation of [^11^C]AZ683—A Novel PET Radiotracer for Colony Stimulating Factor 1 Receptor (CSF1R)

**DOI:** 10.3390/ph11040136

**Published:** 2018-12-13

**Authors:** Sean S. Tanzey, Xia Shao, Jenelle Stauff, Janna Arteaga, Phillip Sherman, Peter J. H. Scott, Andrew V. Mossine

**Affiliations:** 1Department of Medicinal Chemistry, University of Michigan, Ann Arbor, MI 48109, USA; tanzeys@umich.edu; 2Department of Radiology, University of Michigan, Ann Arbor, MI 48109, USA; xshao@umich.edu (X.S.); jrstauff@umich.edu (J.S.); jannaa@umich.edu (J.A.); psherman@umich.edu (P.S.)

**Keywords:** neuroinflammation, microglia, carbon-11, radiochemistry, positron emission tomography

## Abstract

Positron emission tomography (PET) imaging of Colony Stimulating Factor 1 Receptor (CSF1R) is a new strategy for quantifying both neuroinflammation and inflammation in the periphery since CSF1R is expressed on microglia and macrophages. AZ683 has high affinity for CSF1R (K*_i_* = 8 nM; IC_50_ = 6 nM) and >250-fold selectivity over 95 other kinases. In this paper, we report the radiosynthesis of [^11^C]AZ683 and initial evaluation of its use in CSF1R PET. [^11^C]AZ683 was synthesized by ^11^C-methylation of the desmethyl precursor with [^11^C]MeOTf in 3.0% non-corrected activity yield (based upon [^11^C]MeOTf), >99% radiochemical purity and high molar activity. Preliminary PET imaging with [^11^C]AZ683 revealed low brain uptake in rodents and nonhuman primates, suggesting that imaging neuroinflammation could be challenging but that the radiopharmaceutical could still be useful for peripheral imaging of inflammation.

## 1. Introduction

Colony Stimulating Factor 1 Receptor (CSF1R, M-CSF, or cFMS) is a class III receptor tyrosine kinase [1] that regulates immune response by controlling the survival and activity of macrophages and macrophage-like cells [2]. Aberrant activity of CSF1R, or its endogenous ligands (CSF1 and IL-34), plays a role in many disorders that have an immune/inflammatory component [3]. Specifically, chronic inflammation caused by increased activity of macrophages due to increased CSF1R response is present in many autoimmune disorders such as rheumatoid arthritis, inflammatory bowel disease, and autoimmune nephritis, among others [4,5]. The contribution of CSF1R to symptomatic Alzheimer’s Disease (AD) is also well known, due in part to its proliferative effects on microglia, which are associated with neuroinflammation, a hallmark clinical symptom of AD [6,7]. A mechanism for CSF1R involvement in inter-neuronal transmission of pathogenic tau protein by microglia was also recently elucidated [8]. Involvement of CSF1R in certain types of cancers, such as gliomas, also correlates with poor disease prognosis, as proliferation of CSF1R-controlled tumor-associated macrophages (TAMs) correlates with tumor angiogenesis and metastasis [4,9,10,11]. 

Consequently, CSF1R inhibitors (both small molecules and biologics) have been proposed as a means of controlling inflammation in this multitude of diseases and disorders via macrophage depletion/regulation [12]. Many CSF1R inhibitors, including some that are kinase-specific, can be found in both academic and patent literature [4,5,13], and several have proceeded to clinical trials for the treatment of RA [14] and various types of cancer [11]. However, not all macrophage populations are CSF1R-sensitive, necessitating that CSF1R involvement must be positively identified prior to the start of treatment. As CSF1R is a cell surface receptor, its upregulation is only present at the site of inflammation. Although blood biomarkers can be used to directly measure CSF1R involvement in certain diseases, such as lymphoma [11], methods of determining CSF1R involvement and quantifying CSF1R levels at loci not directly connected to the central circulatory system is difficult, particularly in the CNS, and employs either indirect means (i.e., measurement of CSF1 levels as a proxy for CSF1R [14]) or invasive procedures (i.e., immunohistochemistry using a biopsy sample or surgically excised tissue [15,16]). In fact, despite the implication of irregular CSF1R levels in numerous diseases [4], quantitative information on expression levels in disease is generally lacking from the literature. In part this is because CSF1R levels are constantly changing as, for example, macrophages are produced and/or cleared, but also because there is currently an unmet need for a non-invasive method that can positively identify and quantify CSF1R involvement in disease. This goal can be readily achieved with positron emission tomography (PET) imaging, wherein a CSF1R-selective radiolabeled ligand (radiopharmaceutical) would be used to detect changes in activity, expression levels, and localization of CSF1R in a minimally-invasive manner. Furthermore, a brain-penetrant CSF1R-selective PET imaging agent could be used to selectively image microglia, as they are the only cells in the brain that express CSF1R under normal conditions [17]. Microglia associate with amyloid beta plaques in the brain [18], and are implicated in pathological tau transmission [8] and neuroinflammation [6,7], both of which are important in the progression of neurodegenerative disorders.

The current method of choice for imaging macrophages/microglia is by targeting the translocator protein 18 kDa (TSPO). However, TSPO is not an ideal imaging target since it is expressed in various tissue types (in addition to immune cells). Moreover, a single nucleotide polymorphism (SNP) in the TSPO gene has been identified that leads to considerable variability in its expression levels between patients and, consequently, variability in the corresponding PET data [19]. Therefore, a CSF1R imaging agent selective for microglia is of considerable interest for using PET both to quantify CSF1R and as a surrogate biomarker for neuroinflammation (and peripheral inflammation).

Radiopharmaceuticals used in PET imaging are often structural analogs of existing pharmaceutical agents that have been labeled with a positron-emitting radionuclide such as ^11^C, ^18^F or ^124^I. As such, the radiopharmaceutical can be expected to possess the same pharmacokinetic properties as its nonradioactive counterpart and behave accordingly in vivo. Fortunately, lead identification for CSF1R PET radiopharmaceutical development is relatively straightforward because recent interest in developing CSF1R inhibitors has led to hundreds of active compounds, several of which have also been translated into clinical trials (see Figure 1 for several leads) [4,5,13]. PET imaging agents for CSF1R have been reported [20,21], but none have seen widespread use to date. One is a mixed inhibitor of both CSF1R and tropomyosin receptor kinases B and C (Trk B/C) [20], while the second ([^11^C]JHU11744) has shown promise in preliminary evaluation in rodent models of AD and neuroinflammation [21]. 

PET imaging of CSF1R therefore remains underdeveloped and herein we attempt to address this issue through development of [^11^C]AZ683, a new radiopharmaceutical for CSF1R. AZ683 (Figure 1) was selected because it has >250-fold selectivity for CSF1R over 95 other kinases, low plasma protein binding, a good pharmacokinetic (PK) profile, and both fluorine and *N*-methyl moieties which are potential sites for radiolabeling with ^18^F or ^11^C, respectively [22,23,24]. Moreover, AZ683 has low nanomolar affinity for CSF1R (K*_i_* = 8 nM; IC_50_ = 6 nM), making it ideal for PET studies which typically utilize nanomoles-picomoles of radiotracer, and the cLogP of the neutral (uncharged) compound is 3.1 which suggests that it should cross the BBB. Since *N*-methylation of the desmethyl precursor with [^11^C]MeI (or [^11^C]MeOTf) was envisioned to be simpler than ^18^F-labeling of this scaffold, the synthesis and carbon-11 radiolabeling of [^11^C]AZ683 was undertaken for initial evaluation and is described herein. We also report preliminary evaluation of the radiotracer as a CSF1R imaging agent in rodent and non-human primate PET imaging studies.

## 2. Results and Discussion

### 2.1. Synthesis of Reference Standard and Precursor

AZ683 reference standard **6a** and *N*-desmethyl precursor **7** were synthesized via modified literature procedures in five and six steps, respectively (Scheme 1) [23]. Both syntheses diverged from a common intermediate **4**. This common intermediate was synthesized via condensation of 4-bromo-3-ethoxyaniline (**1**) with diethylethoxymethylenemalonate to yield **2**. This was followed by cyclization/chlorination with POCl_3_ and tetrabutylammonium chloride to form chloroquinoline **3**. A subsequent S_N_Ar reaction with 2,4-difluoroaniline yielded intermediate **4**. Buchwald–Hartwig cross-coupling was then used to couple either *N*-Boc piperazine or *N*-methylpiperazine with **4**, yielding intermediates **5a** and **5b** for the reference standard and precursor, respectively. Amidation of the ethyl ester of **5** was performed using formamide/NaOEt to generate reference standard **6a** and *N*-Boc protected precursor **6b**. Final deprotection of the Boc group of **6b** with trimethylsilyl chloride in methanol furnished precursor **7**. Precursor **7** and reference standard **6a** were readily separable on analytical and semipreparative Phenomenex Luna C18 columns using a 30% ethanolic eluent buffered with sodium phosphate at a pH of 6.6 (see *Materials and Methods* for details).

### 2.2. Radiosynthesis of [^11^C]AZ683

Radiolabeling of [^11^C]AZ683 was accomplished by treating precursor **7** with [^11^C]MeOTf (Scheme 2). The labeling reaction was automated using a TRACERLab FX_C-pro_ synthesis module and our standard carbon-11 procedures [25]. Following radiolabeling, [^11^C]AZ683 was purified within the synthesis module via semipreparative HPLC and formulated for injection (0.9% saline solution containing 10% ethanol) using a Waters C18 1cc vac cartridge to trap/release the product. This resulted in an overall non-decay corrected activity yield of 1125 ± 229 MBq (3.0% based upon 37 GBq of [^11^C]MeOTf), radiochemical purity >99%, and molar activity of 153 ± 38 GBq/µmol (n = 4), confirming doses were suitable for preclinical evaluation.

### 2.3. Preclinical PET Imaging

Initial evaluation of the imaging properties of [^11^C]AZ683 was undertaken in female Sprague–Dawley rat. [^11^C]AZ683 was administered via intravenous tail vein injection and rodent brain imaging was conducted for 60 min. To our surprise, [^11^C]AZ683 showed very little brain uptake but did show high uptake and retention in the pituitary and thyroid glands (Figure 2, left). Although both glands are known for expression of CSF1R protein (thyroid) and CSF1R RNA (thyroid and pituitary) [26], we assume the very high uptake is more likely indicative of non-specific binding associated with the lipophilic nature of the compound (Table 1). Since inter-species differences are sometimes apparent between rodents and non-human primates due to the higher metabolic rate in rodents and differing BBB efflux systems, imaging in rhesus macaque brain was also performed. The primate imaging results largely mirrored the rat data, with fairly poor brain influx during the early frames, followed by almost complete washout and little brain retention in a normal brain (Figure 2, right). There was some retention in the central region of the brain that was likely ventricular uptake and, as before, the pituitary gland could be observed in frame and showed a much greater degree of uptake than brain. Overall though, brain uptake in monkey was higher than in rat and there was perhaps some focal uptake in the monkey cerebellum (standardized uptake value (SUV) ~0.3–0.4 at late time points). Given that the cerebellum is an area of known CSF1R expression in humans [26], and CSF1R function is thought to be conserved between vertebrates [27], this signal could correspond to CSF1R, presumably associated with microglia found in the monkey cerebellum [28]. However, this will need to be confirmed in future in vitro experiments with primate brain sections. Target receptor density of CSF1R could ostensibly be low in a non-diseased control animal and would explain poor brain retention, but again normal CSF1R levels are challenging to quantify in vivo as they are transient and expected to fluctuate with the turnover of macrophages and microglia. However, low receptor density would not limit first pass brain influx and efflux which was also quite low. Overall these PET imaging data suggest imaging CSF1R associated with neuroinflammation using [^11^C]AZ683 may be challenging, but that uptake in monkey could be sufficient to observe accumulation in a brain inflammation model. There is literature precedent for TSPO radiotracers with low brain uptake being used to successfully image inflammation in rat models [29,30]. Moreover, the present studies do not rule out labeling the scaffold with a longer-lived PET radionuclide (e.g., ^18^F or ^124^I) and using a prolonged infusion protocol so that sufficient radiotracer accumulates at sites of inflammation. [^11^C]AZ683 could also possibly be used for imaging of peripheral CSF1R to evaluate its role in inflammation outside of the brain.

Given that [^11^C]AZ683 possesses properties mostly consistent with BBB permeability (Table 1) [23,31,32], the lack of brain uptake was unexpected and the reasons for it are unclear. It is possible that [^11^C]AZ683 is a substrate for an efflux transporter on the BBB and since, for example, P-glycoprotein transporter (P-gp) expression is higher in rodents than monkeys (and humans) [33], this could explain the 2–3-fold higher uptake of the radiotracer observed in monkey brain. Given the differences in type and expression levels of efflux transporters between species, monkeys are better for predicting the role of P-gp in limiting brain penetration of drugs in humans [33]. However, as we take a conservative view towards primate safety, methods to determine whether efflux activity is responsible for the low brain uptake of [^11^C]AZ683 (e.g., cyclosporin A blockade of the P-gp transporter [34]) have not been pursued at this time. Alternatively, in this case, cLogP estimates (Table 1) may not be a good indicator of BBB permeability. [^11^C]AZ683 has multiple groups containing nitrogen and oxygen atoms which are ionizable, corresponding to multiple pKa values (Figure 3 [32]). We do not expect the primary amide to limit BBB permeability since we conduct brain imaging with other primary amide-containing radiopharmaceuticals such as [^11^C]LY2795050 [35]. Understanding the relationship between cLogP of charged species as a function of pH is complicated [31], but it is likely that AZ683 is charged at physiological pH and this could be the reason for poor brain uptake. The oxygen and nitrogen atoms in question also participate in hydrogen bonding, and cLogP—the total number of oxygen and nitrogen atoms in a drug molecule (N + O) offers information about logBBB. If cLogP—(N + O) >0, logBBB is likely to be positive and the drug has a good probability of entering the CNS [31]. In the case of AZ863, cLogP—(N + O) = −4, suggesting the number of oxygen and nitrogen atoms may be too high for good CNS penetration. All of these issues should be considered in the design of next generation CSF1R radiopharmaceuticals going forward.

## 3. Materials and Methods

### 3.1. Synthesis

#### 3.1.1. General Considerations

All the chemicals were purchased from commercially available suppliers and used without purification. Automated flash chromatography was performed with Biotage Isolera Prime system. High-performance liquid chromatography (HPLC) was performed using a Shimadzu LC-2010A HT system. ^1^H NMR spectra were acquired using a Varian 400 apparatus (400 MHz) in CDCl_3_ or CD_3_OD. δ are reported in ppm relative to tetramethylsilane (δ = 0), *J* are given in Hz. Mass spectra were measured on an Agilent Technologies (Santa Clara CA, USA) Q-TOF HPLC-MS or Micromass (Manchester, UK) VG 70-250-S Magnetic sector mass spectrometer employing the electrospray ionization (ESI) method.

#### 3.1.2. Compounds Synthesized

*Diethyl 2-(((4-bromo-3-ethoxyphenyl)amino)methylene)malonate* (**2**). To a solution mixture of 4-bromo-3-ethoxyaniline hydrochloride (**1**) (0.66 g, 3 mmol) and K_2_CO_3_ (1.68 g, 12.2 mmol) in MeCN (30 mL) was added diethyl-ethoxymethylene malonate (620 μL, 3 mmol). The reaction was heated to reflux and allowed to stir for 36 h, at which time it was cooled and vacuum filtrated through celite to remove potassium carbonate. The filtrate was purified by flash chromatography using a hexane-EtOAc gradient to yield **2** (0.84 g, 71%). ^1^H-NMR (400 MHz, CDCl_3_) δ 11.00 (d, *J* = 13.5 Hz, 1H), 8.45 (d, *J* = 13.5 Hz, 1H), 7.49 (d, *J* = 8.5 Hz, 1H), 6.63 (d, *J* = 8.5 Hz, 1H), 6.59 (d, *J* = 2.4 Hz, 1H), 4.33–4.22 (m, 4H), 4.09 (q, *J* = 7.0 Hz, 2H), 1.49 (t, *J* = 7.0 Hz, 3H), 1.38–1.31 (m, 6H). [M + H]^+^: Expected 386.0598, Found 386.0604.

*Ethyl 6-bromo-4-chloro-7-ethoxyquinoline-3-carboxylate* (**3**). Compound **2** (0.84 g, 2.2 mmol) was dissolved in dry Toluene (2.5 mL). *Tert*-Butyl ammonium chloride (TBACl: 1.94 g, 7 mmol) was added, followed by POCl_3_ (2 mL, 22 mmol) while stirring at room temperature for 5 min. The reaction mixture was then heated to reflux and stirred for 68 h. After this time the reaction was cooled, diluted with DCM (30 mL) and quenched with water (30 mL). The aq. layer was extracted with further DCM (30 mL) and the combined organic fractions were washed with brine (60 mL), dried (Na_2_SO_4_) and concentrated. The crude material was purified by flash chromatography using a hexane/EtOAc gradient to yield **3** (0.15 g, 17%). ^1^H-NMR (400 MHz, CDCl_3_) δ 9.16 (s, 1H), 8.61 (s, 1H), 7.42 (s, 1H), 4.48 (q, *J* = 7.2 Hz, 2H), 4.28 (dd, *J* = 14.1, 7.0 Hz, 2H), 1.58 (t, *J* = 7.0 Hz, 3H), 1.45 (t, *J* = 7.2 Hz, 3H). [M + H]^+^: Expected 357.9840, Found 359.9820. Cl-37 accounts for difference in expected value.

*Ethyl 6-bromo-4-((2,4-difluorophenyl)amino)-7-ethoxyquinoline-3-carboxylate* (**4**). Compound **3** (0.15 g, 0.41 mmol) was dissolved in ethanol (10 mL). 20 mol % acetic acid (4.7 μL, 0.082 mmol) was added followed by 2,4-difluoroaniline (46 μL, 0.45 mmol, 1.1 eq.). The reaction was heated to reflux and stirred for 24 h. After this time the reaction was cooled and Et_3_N (100 μL) was added to neutralize acetic acid. The crude reaction mixture was purified by flash chromatography to yield title compound **4** (0.11 g, 66%). ^1^H-NMR (400 MHz, CDCl_3_) δ 10.30 (s, 1H), 9.19 (s, 1H), 7.73 (s, 1H), 7.03 (s, 1H), 6.97–6.94 (m, 1H), 6.94–6.92 (m, 1H), 6.84 (t, *J* = 8.3 Hz, 1H), 4.43 (q, *J* = 7.1 Hz, 2H), 4.23 (q, *J* = 7.0 Hz, 2H), 1.53 (t, *J* = 7.0 Hz, 3H), 1.45 (t, *J* = 7.1 Hz, 3H). [M + H]^+^: Expected 451.0463, Found 451.0463.

*Ethyl 4-((2,4-difluorophenyl)amino)-7-ethoxy-6-(4-methylpiperazin-1-yl)quinoline-3-carboxylate* (**5a**). Compound **3** (113 mg, 0.251 mmol) was dissolved in dry toluene (10 mL). To this solution, 1-methylpiperizine (33.4 μL, 0.301 mmol, 1.2 eq.) was added. This solution was aspirated with a syringe and added to a mixture of 2.5 mol % Pd_2_(dba)_3_ (5.75 mg, 0.007 mmol), 2.5 mol % BINAP (3.9 mg, 0.007 mmol) and 1.6 eq. of Cs_2_CO_3_ (0.13 g, 0.402 mmol) under Ar. The reaction was heated to 100°C and stirred for 60 h. After this time the reaction was cooled and quenched with satd. KHCO_3_ (20 mL). The organic layer was washed with water (50 mL) and the water was extracted twice with EtOAc. The organic layers were combined, washed with brine (60 mL), concentrated and dried (Na_2_SO_4_). The residue was purified by flash chromatography using an EtOAc/MeOH gradient to yield compound **5a** (64 mg, 54%). ^1^H-NMR (400 MHz, CDCl_3_) δ 10.14 (s, 1H), 9.11 (s, 1H), 7.32 (s, 1H), 7.26 (s, 1H), 7.00–6.90 (m, 1H), 6.89 (s, 1H), 6.75 (t, *J* = 8.4 Hz, 1H), 4.42 (q, *J* = 7.1 Hz, 2H), 4.21 (q, *J* = 6.9 Hz, 2H), 2.83 (m, 4H), 2.55 (m, 4H), 2.30 (s, 3H), 1.51 (t, *J* = 6.9 Hz, 3H), 1.48–1.44 (m, 3H). [M + H]^+^: Expected 471.2202, Found 471.2202.

*Ethyl 6-(4-(tert-butoxycarbonyl)piperazin-1-yl)-4-((2,4-difluorophenyl)amino)-7-ethoxyquinoline -3-carboxylate* (**5b**). The same procedure described for the synthesis of **5a** was also used to prepare **5b** (61 mg, 44.5%). ^1^H-NMR (400 MHz, CDCl_3_) δ 10.08 (s, 1H), 9.12 (s, 1H), 7.30 (s, 1H), 6.96–6.88 (m, 2H), 6.86 (s, 1H), 6.75 (t, *J* = 8.2 Hz, 1H), 4.43 (q, *J* = 7.1 Hz, 3H), 4.21 (q, *J* = 7.0 Hz, 2H), 3.53–3.45 (m, 4H), 2.75–2.67 (m, 4H), 1.52 (t, *J* = 7.0 Hz, 3H), 1.47 (s, 9H), 1.44 (d, *J* = 7.1 Hz, 3H). [M + H]^+^: Expected 557.2571, Found 557.2571.

*4-((2.,4-Difluorophenyl)amino)-7-ethoxy-6-(4-methylpiperazin-1-yl)quinoline-3-carboxamide* (AZ683 Reference Standard **6a**). Compound **5a** (53 mg, 0.113 mmol) was dissolved in THF (0.1 mL) and formamide (22.4 µL, 0.563 mmol, 5 eq.) was added. To this solution, a 21% wt solution of NaOEt in EtOH (231 µL, 0.563 mmol, 5 eq.) was added. The reaction was heated to reflux and stirred for 16 h, after which time it was cooled and quenched with NH_4_Cl (53 mg, 1 mmol). The reaction mixture was concentrated onto silica and purified by flash chromatography using an EtOAc/MeOH gradient to yield compound **6a** (16 mg, 32%). ^1^H-NMR (400 MHz, CD_3_OD) δ 8.78 (s, 1H), 7.27 (s, 1H), 7.16–7.06 (m, 3H), 6.95 (t, *J* = 8.6 Hz, 1H), 4.25 (q, *J* = 6.9 Hz, 3H), 3.10 (m, 4H), 2.73 (m, 4H), 1.96 (s, 3H), 1.51 (t, *J* = 6.9 Hz, 3H). [M + H]^+^: Expected 442.2049, Found 442.2048.

*tert-Butyl 4-(3-carbamoyl-4-((2,4-difluorophenyl)amino)-7-ethoxyquinolin-6-yl)piperazine-1-carboxylate* (**6b**). The same procedure described for the synthesis of **6a** was used to prepare **6b** (18 mg, 30%). ^1^H-NMR (400 MHz, CDCl_3_) δ 10.44 (s, 1H), 8.76 (s, 1H), 7.27 (s, 1H), 6.94–6.88 (m, 2H), 6.87 (s, 1H), 6.74 (t, *J* = 8.3 Hz, 1H), 6.21 (br. s, 1H), 4.20 (q, *J* = 6.9 Hz, 2H), 3.48–3.47 (m, 4H), 2.73–2.71 (m, 4H), 1.51 (t, *J* = 6.9 Hz, 3H), 1.47 (s, 9H). [M + H]^+^: Expected 528.2417, Found 528.2419.

*4-((2,4-Difluorophenyl)amino)-7-ethoxy-6-(piperazin-1-yl)quinoline-3-carboxamide* (**7**). Compound **6b** (18 mg, 0.034 mmol) was dissolved in dry MeOH (5 mL) and cooled to -78 °C for 5 min. Trimethylsilyl chloride (TMS-Cl, 43.3 µL, 0.341 mmol, 10 eq.) was added and the reaction was allowed to warm up to room temperature and was stirred until deprotection was complete as determined by TLC (~25 hours). The reaction was quenched with water and concentrated to remove solvent and excess TMS-Cl. The concentrate was re-dissolved in MeOH and re-concentrated two more times to ensure complete removal of TMS-Cl. The product was further dried in a vacuum desiccator to yield compound **7** (15 mg, 100%). ^1^H-NMR (400 MHz, CD_3_OD) δ 8.80 (s, 1H), 7.56–7.50 (m, 1H), 7.38 (s, 1H), 7.34 (s, 1H), 7.26–7.18 (m, 1H), 7.14 (t, *J* = 8.4 Hz, 1H), 4.33 (q, *J* = 7.0 Hz, 2H), 3.39–3.33 (m, 4H), 3.22–3.20 (m, 4H), 1.55 (t, *J* = 7.0 Hz, 3H). [M + H]^+^: Expected 428.1893, Found 428.1892.

### 3.2. Radiochemistry

#### 3.2.1. General Considerations

All the chemicals (except for reference standard **6a** and precursor **7** noted above) were purchased from commercially available suppliers and used without purification: sodium chloride, 0.9% USP and Sterile Water for Injection, USP were purchased from Hospira; Dehydrated Alcohol for Injection, USP was obtained from Akorn Inc. (Lake Forest IL, USA) HPLC was performed using a Shimadzu (Kyoto, Japan) LC-2010A HT system equipped with a Bioscan B-FC-1000 radiation detector, and HPLC columns were acquired from Phenomenex (Torrance CA, USA). Other synthesis components were obtained as follows: sterile filters were acquired from MilliporeSigma (Burlington MA, USA); C18 Vac 1cc Sep-Paks were purchased from Waters Corporation (Milford MA, USA); Sep-Paks were flushed with 5 mL of ethanol followed by 10 mL of sterile water prior to use.

#### 3.2.2. Radiosynthesis of [^11^C]AZ683

[^11^C]CO_2_ was produced with a General Electric Healthcare (GE, Uppsala, Sweden) PETTrace cyclotron via the ^14^N(p,α)^11^C reaction. High purity N_2_ (g) containing 0.5% O_2_ was irradiated at 40 µA for 30 min to generate [^11^C]CO_2_ (~111 GBq), which was delivered to a GE TRACERLab FX_C-Pro_ synthesis module and converted to [^11^C]MeOTf (~37 GBq) as previously described [25]. [^11^C]MeOTf was bubbled at 15 mL/min through a solution of precursor **7** (1 mg) in DMF (100 µL) at room temperature for 3 min. Following radiolabeling, the reaction mixture was diluted with HPLC mobile phase and purified by semipreparative HPLC (column: Phenomenex Luna C18, 10µ, 10 × 250 mm; mobile phase: 27% ethanol, 10 mM Na_2_HPO_4_, pH = 5.75; flow rate: 5 mL/min; see Figure 4 for a representative semipreparative HPLC trace). The peak corresponding to [^11^C]AZ683 (t_R_ ~12–14 min) was collected, diluted in water (50 mL), and the resulting solution was passed through a Waters C18 1cc vac cartridge to trap the product. [^11^C]AZ683 was eluted from the cartridge with ethanol (1 mL) and diluted with 0.9% saline solution (9 mL) to provide the formulated product in 10% EtOH. The dose was passed through a 0.22 µm sterile filter into a sterile dose vial. The overall non-decay corrected activity yield of [^11^C]AZ683 was 1125 ± 229 MBq (3.0% based upon 37 GBq of [^11^C]MeOTf) and quality control testing (see below) confirmed radiochemical purity >99%, and molar activity of 153 ± 38 GBq/µmol (n = 4), confirming doses were suitable for preclinical evaluation.

#### 3.2.3. Quality Control Testing of [^11^C]AZ683

##### Visual inspection 

Doses were visually examined and required to be clear, colorless, and free of particulate matter. 

##### Dose pH 

The pH of the doses was determined by applying a small amount of the dose to pH-indicator strips and determined by visual comparison to the scale provided. pH needs to be between 4.5 and 7.5, and the pH of each [^11^C]AZ683 dose synthesized in this study was 5.0.

##### Analytical HPLC

Analytical HPLC was performed using a Shimadzu LC-2010A HT system equipped with a Bioscan B-FC-1000 radiation detector (column: Phenomenex Luna C18, 5µ, 4.6 × 150 mm; mobile phase: 27% ethanol, 10 mM Na_2_HPO_4_, pH: 5.75; flow rate: 0.75 mL/min). Analysis confirmed radiochemical purity >99% (t_R_ of [^11^C]AZ683 ~6 min; see Figure 5 for a typical analytical HPLC trace) and coinjection with unlabeled reference standard **6a** confirmed radiochemical identity (see Figure 6 for a coinjection HPLC trace).

### 3.3. Preclinical PET Imaging

#### 3.3.1. General Considerations 

Rodent and primate imaging studies were performed at the University of Michigan (UM) using a Concorde (CTI-Concorde, Knoxville TN, USA) MicroPET P4 scanner. The University of Michigan is accredited by the Council on Accreditation of the Association for Assessment and Accreditation of Laboratory Animal Care (AAALAC International, Frederick MD, USA) and imaging studies were conducted in accordance with the standards set by the Institutional Animal Care and Use Committee (IACUC) at the University of Michigan (PRO00008103: Biodistribution and Pharmacokinetics of Radiolabeled Compounds; Approval date: 1/16/2018).

#### 3.3.2. Animal Husbandry and Housing

Husbandry and housing for rodents and primates is provided by the University Laboratory for Animal Medicine (ULAM) at UM, and animal facilities are in compliance with the regulations defined by the US Department of Agriculture (USDA).

Monkeys: The University of Michigan PET Center has maintained 2 rhesus macaques for ~15 years and the monkeys are individually housed in adjacent steel cages (83.3 cm high × 152.4 cm wide × 78.8 cm deep) equipped with foraging boxes. They are currently housed in adjacent cages as repeated attempts to socially house them in the same cage have been unsuccessful due to aggressive incompatibility. Cages are metal and do contain gridded floors for radiation safety reasons (radioactive waste is contained to the gridded floor and is easier to clean). Temperature and humidity are carefully controlled, and the monkeys are kept on a 12 h light/12 h dark schedule. Monkeys are fed Lab Fiber Plus Monkey Diet (PMI Nutrition Intl. LLC, Shoreview MN, USA) that is supplemented with fresh fruit and vegetables daily. Water and enrichment toys (manipulanda and food-based treats) are available continuously in the home cage.

Rodents: Rats are housed in Allentown #3 micro ventilated cages (27 cm wide × 49 cm deep × 27 cm high, floor area 923 Sq cm) with animal housing densities set by ULAM and the Guide for the Care and Use of Laboratory Animals. Housing is located on ventilated racks with continuous water and air supply exchange. All animals are provided with LabDiet 5LOD as well as enrichment materials, and are on a light schedule of 12 h light/12 h dark.

#### 3.3.3. Rodent Imaging Protocol

Rodent imaging studies were done using a female Sprague–Dawley rat (weight = 237 g, n = 1). The rat was anesthetized (isoflurane), intubated, and positioned in the PET scanner. Following a transmission scan, the animal was injected (via intravenous (*i.v.*) tail vein injection) with [^11^C]AZ683 (14.8 MBq) as a bolus over 1 min, and the brain imaged for 60 min (5 × 1 min frames-2 × 2.5 min frames-2 × 5 min frames-4 × 10 min frames).

#### 3.3.4. Primate Imaging Protocol

Primate imaging studies were done using a mature female rhesus monkey (weight = 9.4 kg, n = 1). The animal was anesthetized in the home cage with ketamine and transported to the PET imaging suite. The monkey was intubated for mechanical ventilation, and anesthesia was continued with isoflurane. Anesthesia was maintained throughout the duration of the PET scan. A venous catheter was inserted into one hind limb and the monkey was placed on the PET gantry with its head secured to prevent motion artifacts. Following a transmission scan, the animal was injected i.v. with [^11^C]AZ683 (145.0 MBq) as a bolus over 1 min, and the brain imaged for 60 min (5 × 2 min frames-4 × 5 min frames-3 × 10 min frames).

#### 3.3.5. PET Image Analysis

Emission data were corrected for attenuation and scatter, and reconstructed using the 3D maximum a priori (3D MAP) method. By using a summed image, regions of interest (ROI) were drawn on multiple planes, and the volumetric ROIs were then applied to the full dynamic data set to generate time-radioactivity curves.

## 4. Conclusions

In conclusion, we have developed a radiosynthesis of [^11^C]AZ683 for PET imaging of CSF1R and neuroinflammation that provides doses suitable for preclinical use. However, preliminary preclinical PET imaging in rodents and nonhuman primates revealed low brain uptake of [^11^C]AZ683. Overall, these PET imaging data suggest imaging CSF1R associated with neuroinflammation using [^11^C]AZ683 could be challenging and emphasize that high affinity, good selectivity, and appropriate drug-like properties do not guarantee that a compound will make a good radiopharmaceutical for in vivo brain PET. Nevertheless, uptake in monkey could be sufficient to observe accumulation in a brain inflammation model. These studies also do not rule out labeling the scaffold with a longer-lived PET radionuclide (e.g., ^18^F or ^124^I) and using a prolonged infusion protocol to ensure that sufficient radiotracer accumulates at sites of inflammation for imaging and quantitation of CSF1R. [^11^C]AZ683 could potentially also be used for imaging of peripheral CSF1R to evaluate its role in inflammation outside the brain. Future evaluation in animal models of inflammation appears warranted.

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
