# Peer review of "Synthesis and Initial In Vivo Evaluation of [11C]AZ683—A Novel PET Radiotracer for Colony Stimulating Factor 1 Receptor (CSF1R)"

_pharmaceuticals, 2018, doi:10.3390/ph11040136_

Round 1

Reviewer 1 Report

The authors selected AZ683, potentially binding to colony stimulation factor 1 receptor tyrosine kinase, after former intensive efforts for pharmacokinetic optimization of a series of amidoquinolines, for first preclinical PET experiments. They describe synthesis and radioactive labelling with 11C. Colony stimulating factor 1 receptor has been reported to be overexpressed in several brain regions during inflammatory diseases, in cancer cells, macrophages and glia cells. The authors could provide the tracer with a specific activity of approximately 166 GBq/ µmol and apply the drug in two pilot PET experiments to a rat and to a monkey. However, the uptake of the tracer was only in the region of pituitary gland at a range of 2.5-3 but absent in other brain regions. The final explanation for the absent cerebral uptake can be according to the authors either a role of AZ 683 as a substrate of a multidrug resistance protein or charge of its functional groups at physiological pH resulting in inability to cross the blood brain barrier.

Table 1, Page 4/5, Line 151: Please, add the source of these data also for H bond donors and acceptors.

The log P of 5 as the limit for structures suitable for labelling of CNS receptors has its origin in the proposals by Lipinski (90ties; predominantly for classical ligands of G-protein coupled receptors) and is associated in this upper range with high non-specific binding; It appears not very sure that it can be a recommendation for a tyrosine kinase receptor interacting with proteins like CSF1 and Il-34. Especially, for non-peptide ligands at peptide receptors have been proposed also optimized logP values < 3. For AZ683 can be found also logP values below 3.

Page 5, Figure 3 shows the pKa values of functional groups of the tracer. Reference 24 describes, additionally, the formula summarizing cLogP and sum of nitrogen and oxygen atoms for calculation of logBBB . What is the opinion of the author on this approach?

Page 5: Figure 2, please, indicate the pituitary gland with an arrow in the image. Could pituitary gland be visualized also with further peptide receptor ligands (references ?)?. Describe, please, the stereotactic position of the brain slice shown. Tell the reader, please, the cause for the easier approach of the tracer to the pituitary gland.

Page 8: nowadays, it is more common to give the radioactivity in Bq than in Ci and specific radioactivity in GBq/µmol. Please change to Bq.

Page 10: some more information on housing conditions of the animals would be useful.

Page 12, line 408: Is it tPSA, like on page 4 , table 1, or is it TSPA, like page 12, reference 25.

A test of the two presumptions on reasons for the cerebral absence of the tracer would be welcome. Finally, the question arises, if the authors can expect an accumulation of the tracer in the brain sufficient for visualization with PET without stimulation of inflammatory processes? Which density of the receptor do the authors expect under control conditions.

Author Response

The authors selected AZ683, potentially binding to colony stimulation factor 1 receptor tyrosine kinase, after former intensive efforts for pharmacokinetic optimization of a series of amidoquinolines, for first preclinical PET experiments. They describe synthesis and radioactive labelling with 11C. Colony stimulating factor 1 receptor has been reported to be overexpressed in several brain regions during inflammatory diseases, in cancer cells, macrophages and glia cells. The authors could provide the tracer with a specific activity of approximately 166 GBq/ µmol and apply the drug in two pilot PET experiments to a rat and to a monkey. However, the uptake of the tracer was only in the region of pituitary gland at a range of 2.5-3 but absent in other brain regions. The final explanation for the absent cerebral uptake can be according to the authors either a role of AZ 683 as a substrate of a multidrug resistance protein or charge of its functional groups at physiological pH resulting in inability to cross the blood brain barrier.

Table 1, Page 4/5, Line 151

Response: The sources are already cited at the top (in the table header)

The log P of 5 as the limit for structures suitable for labelling of CNS receptors has its origin in the proposals by Lipinski (90ties; predominantly for classical ligands of G-protein coupled receptors) and is associated in this upper range with high non-specific binding; It appears not very sure that it can be a recommendation for a tyrosine kinase receptor interacting with proteins like CSF1 and Il-34. Especially, for non-peptide ligands at peptide receptors have been proposed also optimized  logP values < 3. For AZ683 can be found also logP values below 3.

Response: This is correct, and AZ683 does, in fact, fall close to this optimal logP limit. An extra entry has been added to Table 1 to reflect both Lipinski’s Ro5 and this revised logP value (2.7 is the value given in Pajouhesh reference).

Page 5, Figure 3 shows the pKa values of functional groups of the tracer. Reference 24 describes, additionally, the formula summarizing cLogP and sum of nitrogen and oxygen atoms for calculation of logBBB . What is the opinion of the author on this approach?

Response: a discussion of logBBB has been added.

Page 5: Figure 2, please, indicate the pituitary gland with an arrow in the image. Could pituitary gland be visualized also with further peptide receptor ligands (references ?)?. Describe, please, the stereotactic position of the brain slice shown. Tell the reader, please, the cause for the easier approach of the tracer to the pituitary gland.

Response: We have added arrows to the Figure as requested and discussed glandular uptake. We have not correlated the PET images to the stereotactic position since the paper makes general statements about the whole brain and cerebellum only. Given the resolution of the PET scanner, we makes no claims about brain subregions at the stereotactic level. We hope this is acceptable.

Page 8: nowadays, it is more common to give the radioactivity in Bq than in Ci and specific radioactivity in GBq/µmol. Please change to Bq.

Response: Changed as requested.

Page 10: some more information on housing conditions of the animals would be useful.

Response: Details of animal housing have been added to Section 3.3.2.

Page 12 , line 408: Is it tPSA, like on page 4 , table 1, or is it TSPA, like page 12, reference 25

Response: tPSA is correct and this has been amended in Reference 25.

A test of the two presumptions on reasons for the cerebral absence of the tracer would be welcome. Finally, the question arises, if the authors can expect an accumulation of the tracer in the brain sufficient for visualization with PET without stimulation of inflammatory processes? Which density of the receptor do the authors expect under control conditions.

Response: Despite the implication of irregular CSF1R levels in numerous diseases, quantitative information on expression levels in disease is generally lacking from the literature. This is likely because it is transient in nature and fluctuates with turnover of microglia and macrophages (we have been unable to find a literature Bmax value). We have added a comment reflecting this to the manuscript. It’s a big reason why such a radiotracer would be useful to the community.

It is possible that [11C]AZ683 is a substrate for an efflux transporter on the BBB and since, for example, P-glycoprotein transporter (P-gp) expression is higher in rodents than monkeys (and humans), this could explain the 2-3-fold higher uptake of the radiotracer observed in monkey brain. Given the differences in type and expression levels of efflux transporters between species, monkeys are better for predicting the role of P-gp in limiting brain penetration of drugs in humans. However, as we take a conservative view towards primate safety, methods to determine whether efflux activity is responsible for the low brain uptake of [11C]AZ683 (e.g. cyclosporin A blockade of the P-gp transporter) have not been pursued at this time and we hope this is acceptable.

Reviewer 2 Report

The subject of the paper submitted by Tanzey et al. falls within the scope of the journal pharmaceuticals. The paper deals with the synthesis, 11C-labelling and initial preclinical investigation of a new radiotracer for imaging the colony stimulating factor 1 receptor (CSFR1) with PET. This is a very important subject because of the high impact of this target on inflammation in brain and periphery.

Generally, the study is well designed. The provided data are convincing and conclusive. However the interpretation should be revised. The authors are encouraged to provide a more optimistic view on their data.

The following are some comments and corrections that are suggested to be considered by the authors:

Abstract

Line 13: Microglia is only found in brain not in periphery. Please revise.

Introduction

Line 30: Replace “Rheumatoid” by “rheumatoid”.

Results and Discussion

Lines 118-119: Please follow the “Consensus nomenclature rules for radiopharmaceutical chemistry”.

Page 4: It is suggested to replace “very low/poor brain uptake” by “moderate brain uptake”. The uptake in monkey appears to be high enough to expect radiotracer accumulation in a brain inflammation model. There are examples of reasonable radiotracers with only moderate brain uptake (e.g. Theranostics 2015, Vol. 5, Issue 9, page 961, Mol Imaging Biol (2017) 19:77-89). Please discuss accordingly. Others might by found.

Since it is mentioned in the abstract, please include the CSFR1-related aspect of peripheral inflammation also in the discussion based on suitable references.

Page 5 Fig.2: Please indicate in the images where the brain, pituitary and thyroid are found.

Material and Methods

Line 249: Replace “sterile water” by “Sterile Water”.

Author Response

The subject of the paper submitted by Tanzey et al. falls within the scope of the journal pharmaceuticals.The paper deals with the synthesis, 11C-labelling and initial preclinical investigation of a new radiotracer for imaging the colony stimulating factor 1 receptor (CSFR1) with PET. This is a very important subject because of the high impact of this target on inflammation in brain and periphery.

Generally, the study is well designed. The provided data are convincing and conclusive. However the interpretation should be revised. The authors are encouraged to provide a more optimistic view on their data.

The following are some comments and corrections that are suggested to be considered by the authors:

Abstract

Line 13: Microglia is only found in brain not in periphery. Please revise.

Response: We have addressed this by changing “microglia” to “microglia and macrophages”.

Introduction

Line 30: Replace “Rheumatoid” by “rheumatoid”

Response: this has been changed.

Results and Discussion

Lines 118-119: Please follow the “Consensus nomenclature rules for radiopharmaceutical chemistry”

Response: Ci / mCi have been changed to MBq and GBq.

Page 4: It is suggested to replace “very low/poor brain uptake” by “moderate brain uptake”. The uptake in monkey appears to be high enough to expect radiotracer accumulation in a brain inflammation model. There are examples of reasonable radiotracers with only moderate brain uptake (e.g. Theranostics 2015, Vol. 5, Issue 9, page 961, Mol Imaging Biol (2017) 19:77-89). Please discuss accordingly. Others might by found.

Response: Discussion has been updated and citations have been added.

Since it is mentioned in the abstract, please include the CSFR1-related aspect of peripheral inflammation also in the discussion based on suitable references.

Response: Some discussion with supporting references was already included in the introduction and we believe this is adequate:

“Specifically, chronic inflammation caused by increased activity of macrophages due to increased CSF1R response is present in many autoimmune disorders such as rheumatoid arthritis, inflammatory bowel disease, and autoimmune nephritis, among others.4, 5

Page 5 Fig.2: Please indicate in the images where the brain, pituitary and thyroid are found.

Response: labels have been added to the figure.

Material and Methods

Line 249: Replace “sterile water” by “Sterile Water”

Response: changed as requested.

Round 2

Reviewer 1 Report

In reference 26, I cannot recognize data on the expression of CSF1R in cerebellum of monkey. (citation page 5, lines 142-144).

Please, replace it by a reference with the respective content cited on page 5.

Author Response

In reference 26, I cannot recognize data on the expression of CSF1R in cerebellum of monkey. (citation page 5, lines 142-144).

Please, replace it by a reference with the respective content cited on page 5.

Response: Section has been expanded, and additional citations have been added as requested.

Reviewer 2 Report

The manuscript reads very well now. The “Consensus nomenclature rules for radiopharmaceutical chemistry” request the use of "molar activity" instead of "specific activity". Please change accordingly.

Author Response

The manuscript reads very well now. The “Consensus nomenclature rules for radiopharmaceutical chemistry” request the use of "molar activity" instead of "specific activity". Please change accordingly.

Response: Specific activity has been changed to molar activity as requested. We have also changed 3 Ci of CO2 and 1 Ci of MeOTf to GBq in the methods section since we missed those in the last revision.